# Electrical Characterization of the Backside Interface on BSI Global Shutter Pixels with Tungsten-Shield Test Structures on CDTI Process

**DOI:** 10.3390/s20010287

**Published:** 2020-01-04

**Authors:** Célestin Doyen, Stéphane Ricq, Pierre Magnan, Olivier Marcelot, Marios Barlas, Sébastien Place

**Affiliations:** 1STMicroelectonics, 850 rue Jean Monnet, 38920 Crolles, France; marios.barlas@st.com (M.B.); sebastien.place@st.com (S.P.); 2ISAE-SUPAERO, Université de Toulouse, 10 Avenue Edouard Belin, 31055 Toulouse, France; pierre.magnan@isae-supaero.fr (P.M.); olivier.marcelot@isae-supaero.fr (O.M.)

**Keywords:** back-side illumination imager, dark current, Interface states, passivation, electrical device characterization, test structures, W-shield, CDTI, charge pumping, CV

## Abstract

A new methodology is presented using well known electrical characterization techniques on dedicated single devices in order to investigate backside interface contribution to the measured pixel dark current in BSI CMOS image sensors technologies. Extractions of interface states and charges within the dielectric densities are achieved. The results show that, in our case, the density of state is not directly the source of dark current excursions. The quality of the passivation of the backside interface appears to be the key factor. Thanks to the presented new test structures, it has been demonstrated that the backside interface contribution to dark current can be investigated separately from other sources of dark current, such as the frontside interface, DTI (deep trench isolation), etc.

## 1. Introduction

Backside illuminated (BSI) imager technologies are nowadays widely used thanks to their advantages, such as better fill factor and better light collection, compared to frontside technologies. As the frontside interface, the backside interface can be a source of dark current. Key factors are interface states and their passivation, here managed through negatively-charged high-K dielectric to easily accumulate holes and, therefore, passivate the interface [1,2,3,4,5]. Interface state density drives the interface generation, passivation modulates the quantity of electrons that could escape from the interface region and reach photodiode. Therefore, it is crucial to develop this interface characterization. A characterization method named COCOS (corona oxide characterization of semiconductor) [6] already exists which enables the extraction of the density of the interface states and the charges within the oxide just after the deposition of a material. Here a new methodology is presented that enables characterizing the backside interface at the very end of the process and so, in the final pixel environment, which cannot be done by COCOS because of the metal shield preventing the light needed for the measurement from reaching the interface. It is based on well-known and relatively simple electrical characterization techniques applied on new dedicated test structures that benefit from the tungsten (W) layer in the technology for light shielding purposes. Additionally, repeated measurements will be possible with these structures (see Section 4), which cannot be done with COCOS due to the charge injected for the measurement, which cannot be removed correctly. Finally, the test structures are embedded on fully-processed wafers, so wafers are not dedicated only for the extraction of interface states and charge within the oxide, unlike COCOS measurements done on unpatterned wafers. Indeed, multiple structures are also present for different tests important for the development of the technology. Thus, with these test structures, dark current measurements and extractions presented in this paper can be done on the same wafer.

Our use-case detailed in this paper is based on three wafers from BSI CDTI (capacitive deep trench isolation [7]) technology with Ta_2_O_5_/Al_2_O_3_ backside dielectric process variants. For the 31 × 863 array of 3.2 µm × 3.2 µm, covered by W shield pixels, the median dark current (Idark) distributions measured on three wafers are presented in Figure 1. Dark current is measured at 60 °C, and further in the paper the different measurements on the test structures are conducted at 25 °C, which is more convenient. This does not affect the results because the measured parameters do not vary with the temperature. The temperature during measurements is controlled by chillers and temperature controllers. For measurements, the pixels’ CDTI are biased at −2 V, which is their operating voltage, in order to have an accumulation layer at their interface which is, therefore, passivated. The W-shield is not biased because, with negative charge within the high-K, the backside interface should be passivated. Atypical behaviors can be seen on wafers 2 and 3. For both wafers, there is a higher and non-uniform dark current. In order to investigate the root cause of the dark current, two special test structures are measured on the same dies and physically near the pixel array used for Idark extraction (Figure 2) in order to investigate if the Idark excursions come from backside interface and dielectric properties.

In this paper the different test structures are first presented. In Section 2, the density of interface states (Dit) is investigated with a tentative correlation with Idark. A similar exercise is performed on the passivation quality, detailed in Section 3. Finally, a charging phenomenon present on both structures is presented and discussed.

## 2. Test Structures Description

To study the potential correlation between Idark, density of interface states (Dit) and quality of passivation achieved through negative charge density within the oxide Neff, two test structures have been developed in a BSI global shutter pixel technology with CDTI using the W-shield as a gate: a backside MOS capacitor (Figure 3) (L=500 µm, W=31.6 µm) and a backside W-shield gate pseudo-MOS transistor (Figure 4) (L=10 µm, W=27 µm).

The capacitor is a simple metal-dielectric-semiconductor device and is functional (see Figure 5).

In the transistor structure, CDTI are biased in a way that there is always an inversion layer at their interface in order to connect surface source and drain to the bottom interface of interest, that is to say, the channel of the W-shield gate transistor. Figure 6 shows the Id-Vg characteristic of the transistor test structure. To perform this measurement, CDTI are biased at 4 V to be sure to have $V_{CDTI}>V_{TCDTI}$ where $V_{TCDTI}$ is the threshold voltage of the CDTI.

In the W-shield transistor test structure, the channels facing CDTI are in series with the W-shield MOS, one can wonder if $V_{CDTI}$ can have an impact on the measurements done with the transistor test structure. To study this potential effect, Id(Vg) measurements are performed with several $V_{CDTI}$. The results can be seen in Figure 7. These curves show that $V_{CDTI}$ does not impact the MOS threshold voltage (Vt). The subthreshold conduction is also identical for the three biases. Only the conduction part looks different, as a higher $V_{CDTI}$ leads to more current on the drain. To conclude, as long as it is high enough, $V_{CDTI}$ does not seem to affect most characterizations that can be done on this structure and pure backside interface information can be extracted.

## 3. Interface States Characterization

The dark current generated by an interface may be dominated by interface states if their density is high enough. In order to extract the interface states density (Dit) for each die, the charge pumping method [8,9,10,11,12], usually used for classic transistors, is applied on the W-shield transistor test structure (Figure 4). This method consists of applying a pulse to the gate which controls the studied interface. By doing so, the interface is alternatively accumulated and inverted. During transitions the interface states emit and capture electrons which create a pumped current which can be measured though the bulk contact. During accumulation or inversion, there is no pumped current measurable because all interface states are almost all occupied. The implementation of the method is illustrated by Figure 8.

The CDTI are biased at 4 V in order to always have an inversion layer at their interface so, as it is explained above, the CDTI interface does not contribute to the pumped current, only the backside interface is characterized. The drain and the source are grounded. A trapezoidal pulse (Figure 9) with the following characteristics are applied to the W-shield gate: $V_{g\_low}$ is swept from 2 to 12 V, $\Delta V_{gpulse}$ = 6 V, minimum frequency of the signal f = 20 kHz, and rise and fall time $T_{r,f}$ = 1 µs. The pumped current is measured through the bulk contact. The maximum measured pumped current is expected to be proportional to the mean Dit ($cm^{-2}$) [8]:(1)$$I_{{pump}_{max}}=D_{it}qAf$$

Here, $I_{{pump}_{max}}$ represents the maximum pumped current, q is the elementary charge, and A is the gate area. Figure 10 shows the maximum pumped current as a function of the frequency. It can be seen that the maximum measured pumped current is linear with f which is in agreement with Equation (1). The slope extraction gives Dit.

However, some cautions have to be taken. First it must be checked that the device response is not limited by a geometric component [11]. Indeed, the length of the gate is 10 µm which may be favorable for such a phenomenon. Actually, when the interface goes back from inversion to accumulation, electrons flow back to the source and drain. If the gate is too long, some electrons do not have the time to reach the source or the drain before the interface reaches the accumulation and they are recombined. This creates an additional current which does not come from the interface states and the pumped current is overestimated. An easy way to verify the presence of this effect is to perform the experiment on structures with different lengths. If the results depend on the gate length, the extraction of the density of interface states is inaccurate. In this study, only one length is available, so in order to find out if this phenomenon is present or not, the method described in [13] can be used. In this work, the pumped charges ($Q_{CP}=\frac{I_{cp}}{f}$) is plotted as a function of the rise and fall time which are equals. If there is not geometric component, the pumped charges should vary as follow:(2)$$Q_{CP}\propto \ln\left(T_{r,f}\right),$$
where $Q_{CP}$ is the pumped charges (C) and $T_{r,f}$ is the rise and fall time (s). Measurements performed on W-shield test structures are shown in Figure 11. It can be seen that the pumped charge is actually linear with the rise and fall time, which confirms that the measurements are not affected by a geometric component. If it was the case, the measured pumped charge is no longer linear with time and starts to increase for small rise and fall time because, as for a too long gate, electrons do not have time to reach the source and drain and recombine. Here, the loss of the linearity and the decreased of the pumped charge measured for small rise and fall time is due to the fact that transition times are too short and the interface states do not have the time to respond, and so the measured pumped charge starts to fall.

The second caution which has to be taken is the frequency range of the measurements. The pumped charges should be constant with the frequency. However, when the frequency becomes too low, the structure is in inversion or in accumulation for too long time and tunneling effect may occur [14]. If this is the case the measured pumped charge increases because the charge pumping method starts to characterize traps which are within the high-K [14] and not at the interface. At high frequency, only the interface is characterized. Measurements of pumped charge vs. the frequency are shown in Figure 12. The pumped charge is effectively constant with the frequency from 10 kHz. Under this frequency, $Q_{CP}$ starts to increase. To conclude, the measurements have to be done with frequencies higher than 10 kHz.

With the IdVg curves in Section 2, it is important to see if the charge pumping measurements are affected by the CDTI bias. Figure 13 shows the maximum pumped current as a function of the frequency for different CDTI bias. It can be seen that the slopes of the different measurements are the same so the interface state density which is extracted is equal for all CDTI biases. It can be concluded that the CDTI bias does not affect the characterization of the interface state density.

Knowing these cautions, measurements are performed on the three wafers using the sampling map of Figure 2. Figure 14 presents the Idark vs Dit scatter plot for the three wafers. From the scatter plot no clear tendency can be extracted. Idark looks not to be correlated to Dit in this case. Therefore, the measured dark current differences between the three wafers cannot be explained by a difference of Dit, as all the wafers have an equivalent level of interface states ($\sim 5\times 10^{11}/{\mathrm{cm}}^{2}$). This value is adequate with what can be obtained with the COCOS measurement ($\sim 3.5\times 10^{11}/{\mathrm{cm}}^{2}$).

## 4. Quality of the Passivation Characterization

As interface states are not responsible for Idark excursions, the quality of the passivation might be the root cause of the Idark response seen in Figure 1. Indeed, with very similar Dit extracted on the wafers, the field effect passivation, that is, having an electric field at the interface to accumulate charges in order to passivate the interface, can have a key contribution. The electric field is induced by negative charges within the oxide (*N_EFF_*).

For this analysis the Maserjian’s function is used on C-V measurements to estimate the effective charge density within the oxide on the capacitor structure (Figure 3) [15,16]:(3)$$Y\left(V_{g}\right)={\left(\frac{1}{C}\right)}^{3}\frac{dC}{dV_{g}}.$$

Thanks to this function, the substrate doping concentration ($N_{a}$) and the flat band voltage ($V_{FB}$) can be extracted:(4)$$Y_{min}=-\frac{1}{q\varepsilon_{Si}N_{a}}$$
(5)$$Y\left(V_{FB}\right)=-\frac{Y_{min}}{3},$$
where $Y_{min}$ is the minimum reached by the Maserjian’s function and $\varepsilon_{Si}$ the silicon permittivity. Then it is possible to calculate *N_EFF_*:(6)$$N_{EFF}=\frac{-\left(V_{FB}-W_{MS}\right)C_{ox}}{q},$$
where $N_{EFF}$ is the effective number of charges within the oxide, $W_{MS}$ the metal semiconductor work function and $C_{ox}$ the oxide capacitance. For the measurements, the following characteristics are applied: $V_{g}$ is swept from 2 to 15 V, the signal frequency applied is f = 50 kHz, and the modulation of the signal is 0.02 V. An example of an obtained *Y* function for one capacitor structure can be seen in Figure 15. Figure 16 presents the scatter plots of Idark as a function of the extracted *N_EFF_* for the three wafers.

On wafer 1, it was difficult to extract an eventual correlation between Idark and the density of charge within the oxide probably because the Idark variation is weak (Figure 1). However, on wafers 2 and 3, a clear tendency can be identified between *N_EFF_* and Idark. In order to observe if there is a more general tendency, a global scatter plot with the three wafers together is shown in Figure 17.

In this figure, two regions of the scatter plot can be distinguished: a first one where the dark current is almost insensible to *N_EFF_* and a second one where the dark current increases according to the Neff reduction (in absolute value). According to [1], the plateau is explained by the fact that when a certain number of charges within the oxide is reached, the interface is fully passivated and the Idark from the backside interface becomes negligible. Therefore, the measured dark current comes from other parts of the pixel, such as DTI, frontside interface, silicon volume, etc. On the contrary, for the lowest *N_EFF_* values in absolute value, as on some wafer 2 and 3 dies, the passivation is not efficient enough and a significant Idark appears coming from the BS interface. Here, again, the results obtained are consistent with what can be measured with the COCOS technique (same order of magnitude)

These new test structures, backside W-shield gate capacitor and transistor, are functional and enable to use very well-known and quite simple techniques in order to characterize the backside interface of a BSI pixel with W-shield at the end of the process. Thanks to these structures, it is possible to investigate if the measured dark current is coming from the backside interface or not and to determine the possible root causes of this dark current (here, the quality of the passivation). The extractions done with these measurements are in good agreement with what can be obtained with COCOS. To be complete, it can be determined that the structures with the W-shield represent the dark reference shielded pixels very well. Their response can be different in terms of dark current than usual pixels (without the shield). However, this can be controlled by comparing the dark current from usual pixels with the dark reference pixel (shielded).

The next section will present a charging effect that can be observed. This effect does not affect the results presented in this study, but it has to be known in order to perform the best measurements.

## 5. Charging Effect

Measurements on our W-shield MOS or capacitor are performed up to relatively high voltages to achieve extractions. If they are repeated a second time, and more, characteristics look shifted, as illustrated in Figure 18. This is attributed to a charging effect. The same observation can be done on the capacitor structure.

On both structures, the curve shift seems to be related to a $V_{FB}$ shift. During the measurement, $V_{g}$ reaches high voltage (15 V and above) which can induce a negative charge injection within the oxide and results in higher $V_{FB}$ voltage.

In addition to this charging effect, a hysteresis phenomenon may be seen on both structures as well. For example, Figure 19 shows successive IdVg measurements operated on the transistor test structure. $V_{g}$ is swept from 0 to 12 V and then from 12 to 0 V, and finally from 0 to 12 V again. An effect of charging/discharging can be seen. However, after a measurement the curve may come back to the initial state by applying a negative bias of −5 V for 1000 s on the gate. The example of Id(Vg) measurements can be taken. Figure 20 shows the results of this procedure. After making a third measurement, it can be seen that the curve obtained is almost the same as the first one, and the charging effect is recovered.

Some other effects can be seen associated to this charging effect. First, in the case of the charge pumping, between the first measurement and the other ones, the maximum pumped current slightly increases but does not strongly affect the extraction of Dit (Figure 18). In the case of the C-V measurement, Figure 21 shows the hysteresis effect that can be obtained on the capacitor structure.

Here, the different sweeps have been performed between −10 and 20 V. In addition to the hysteresis effect, it can be seen that the bump initially present in the first sweep disappears in the second sweep (from 20 to −10 V). This bump in C-V measurements can be the result of the interface trapped charge [17,18]. When the measurement starts, the structure is in accumulation and holes can be trapped by interface states leading to bump creation. When the high voltage is reached, the interfaces states are discharged, and they capture an electron, in addition to the charging effect, and so the bump disappears. It reappears with the third sweep showing that holes were captured again.

Back to pixels operating conditions, it should be also noted that such conditions are not applied and such effects should not happen. The different extractions presented earlier in this paper were made during the first measurement when the structures are not already stressed. Multiple measurements can be made on these structures since the charging effect can be recovered (unlike COCOS measurements).

## 6. Conclusions

With these MOS capacitor and W-shield gate transistor test structures, it is possible to electrically characterize the backside interface of BSI technology at the end of a process using a tungsten shield. By means of two known characterization methods, Dit and *N_EFF_*, which are the two important parameters for dark current, can be extracted. It is, therefore, possible to investigate if the dark current mainly comes from the backside interface, and to discriminate the origin of the backside dark current. In the case presented in this study, the difference in Idark behavior is explained by quality passivation differences of the backside interface between wafers. COCOS measurements are useful to characterize the interface just after a material deposit, however, it cannot be used with a fully processed wafer, unlike the methodology used on the new structures presented in this study. A drawback of this method is the presence of a charging effect that forces some caution on the execution of measurements, but this effect can be recovered and is not present in pixel operating conditions. In addition to these Idark contribution studies, these dedicated devices with associated characterizations can be helpful for process monitoring, TCAD calibration, and reliability works.

To go further, a comparison with an analytical model of the dark current generated at the backside interface and TCAD simulations are under study to reinforce all the obtained results.

## Figures and Tables

**Figure 1 sensors-20-00287-f001:**
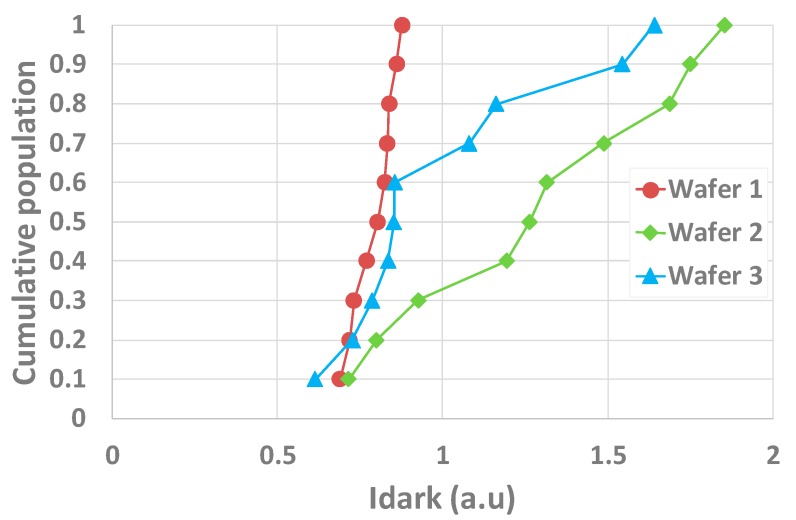
Cumulative population of the pixel dark current measured on test chips on several dies of three wafers showing distinct behaviors depending on the wafer.

**Figure 2 sensors-20-00287-f002:**
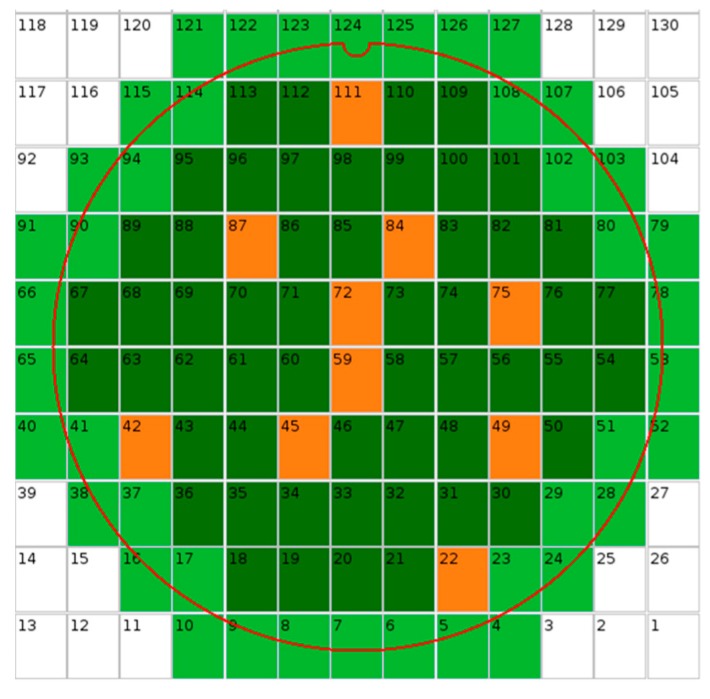
Wafer map showing the dies used for Idark and test structure measurements in orange.

**Figure 3 sensors-20-00287-f003:**
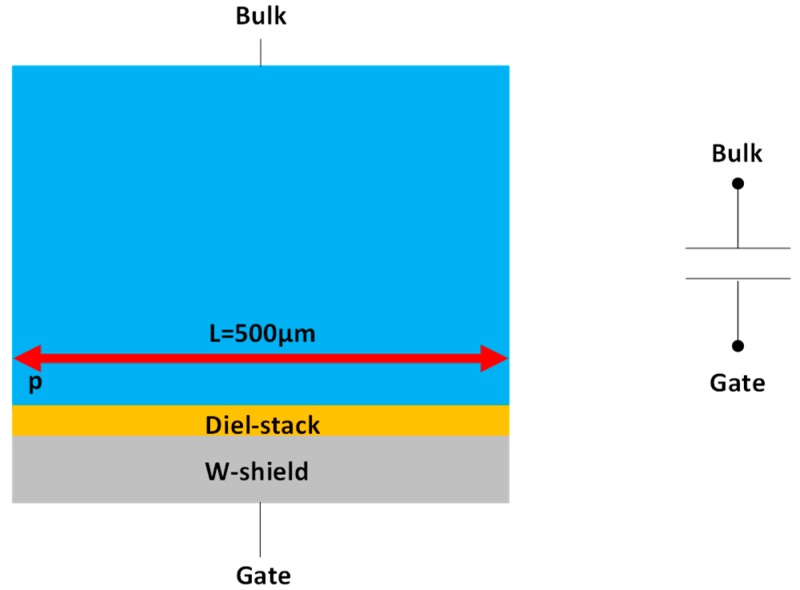
Illustration of the capacitor test structure used to extract $N_{eff}$. It is composed of a W-shield used as a gate, a dielectric stack as the MOS oxide, and a p-doped substrate.

**Figure 4 sensors-20-00287-f004:**
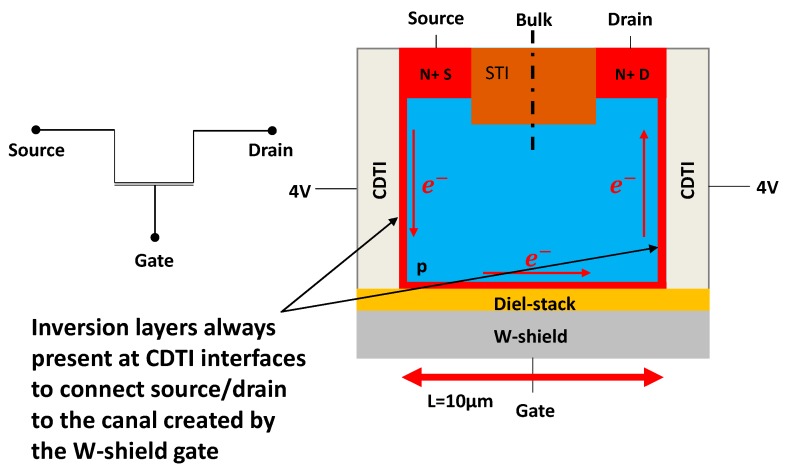
Illustration of the transistor test structure used to extract $D_{it}$. It is composed of a W-shield used as a gate, a dielectric stack as the MOS oxide, and a p-doped substrate. CDTI on both sides are present to connect the source and drain to the surface controlled by the W-shield gate. An STI is present (shallow trench isolation) between the source and drain to prevent punch-through).

**Figure 5 sensors-20-00287-f005:**
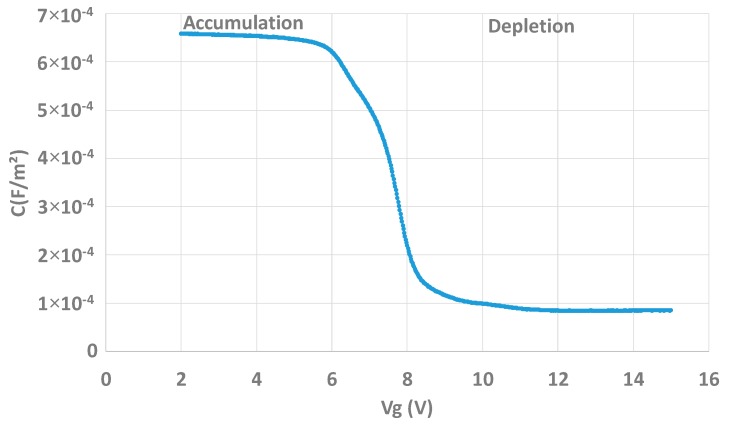
C(Vg) curve showing the conventional operation of the capacitor test structure with accumulation and depletion regimes.

**Figure 6 sensors-20-00287-f006:**
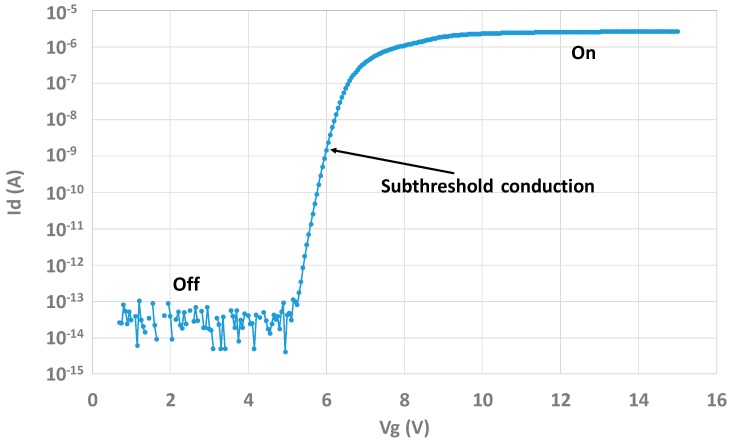
Id(Vg) curves showing the conventional operation of the W-shield gate transistor with clear ON/OFF behaviors.

**Figure 7 sensors-20-00287-f007:**
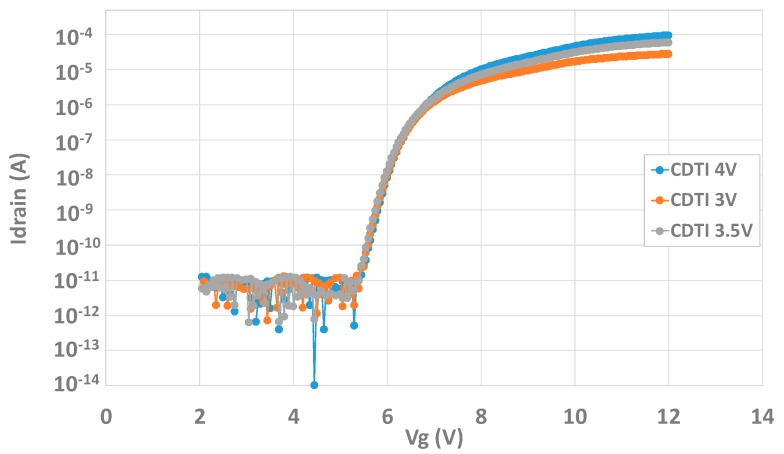
Id(Vg) curve for different VCDTI in order to check the effect of the CDTI bias on the IdVg characteristic.

**Figure 8 sensors-20-00287-f008:**
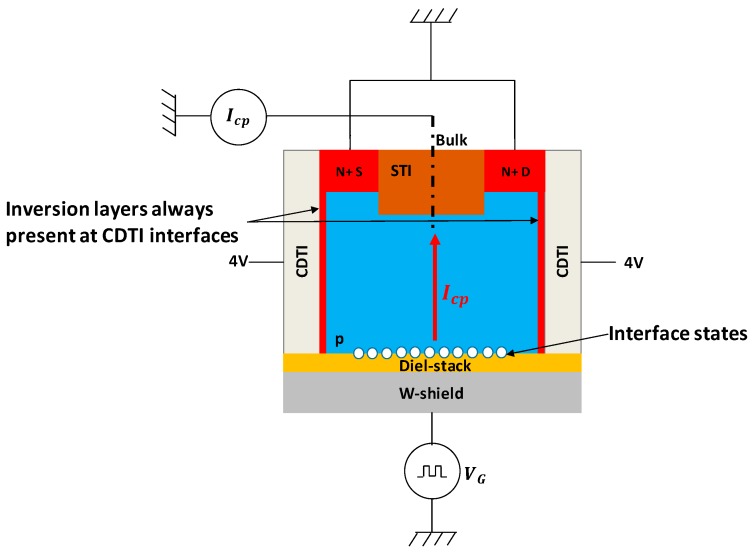
Charge pumping method applied to the W-shield transistor test structure.

**Figure 9 sensors-20-00287-f009:**
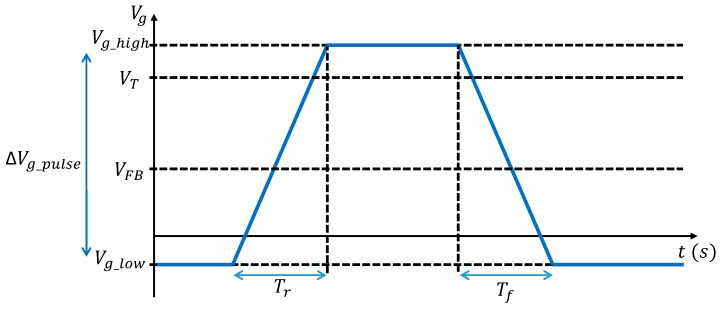
Chronogram of the pulse applied on the gate necessary for the charge pumping method.

**Figure 10 sensors-20-00287-f010:**
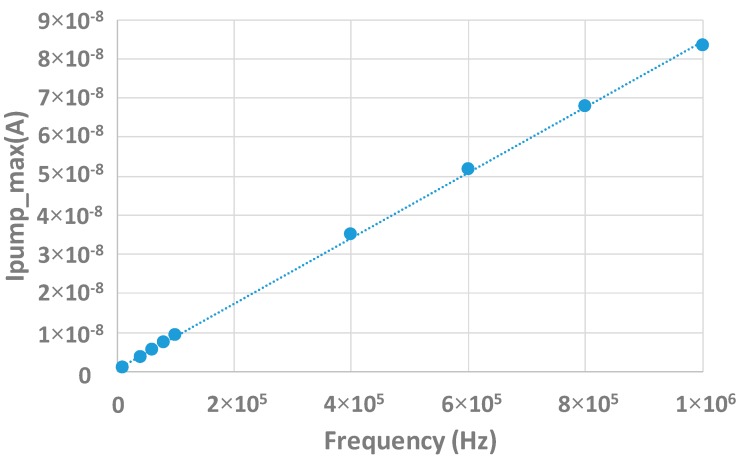
Ipump_max(*f*) measurement for various frequencies.

**Figure 11 sensors-20-00287-f011:**
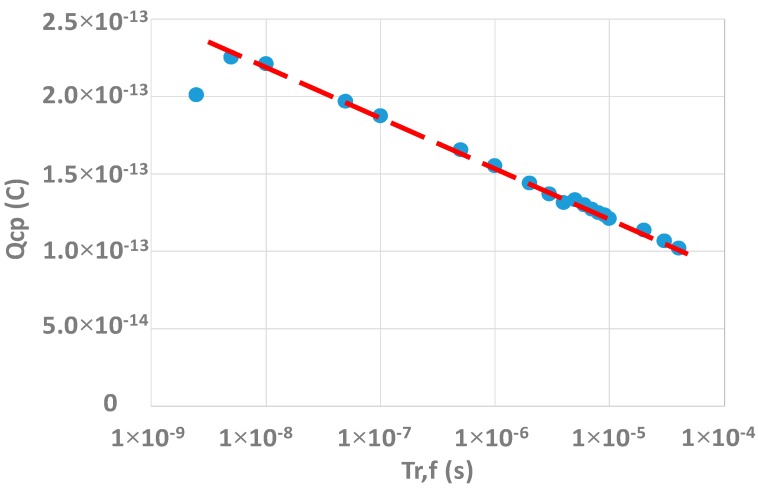
Measurements of pumped charge for various rise and fall time *Qcp* = *f*(*Tr*,*f*).

**Figure 12 sensors-20-00287-f012:**
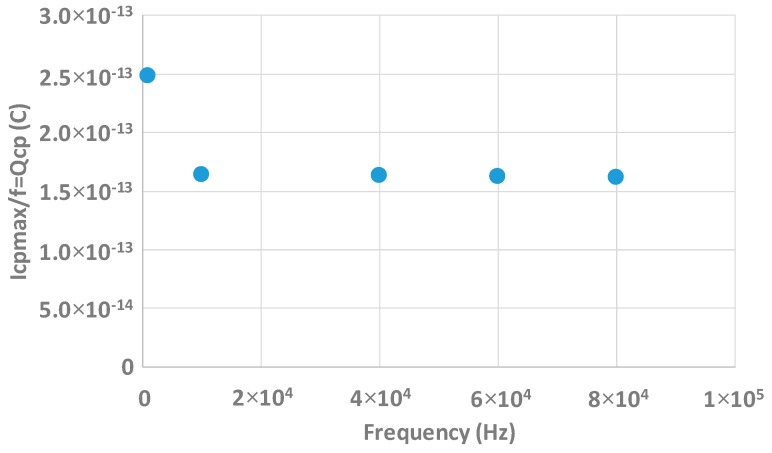
Dependency of the pumped charge in function of the frequency.

**Figure 13 sensors-20-00287-f013:**
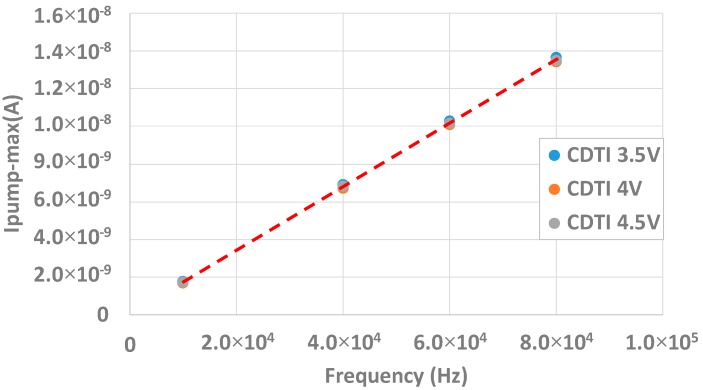
Ipump_max in function of the frequency for several CDTI bias in order to demonstrate that the CDTI voltage does not affect the extraction of the interface state density.

**Figure 14 sensors-20-00287-f014:**
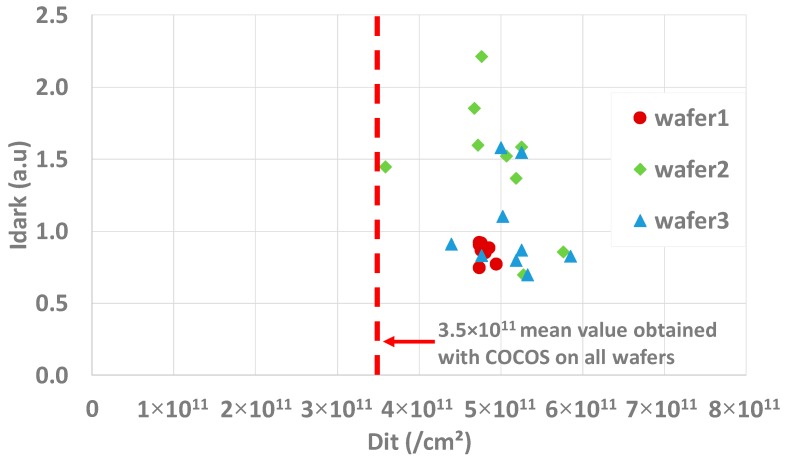
Idark(Dit) scatter plot. All the wafers show a same level of Dit. The mean value obtained by COCOS measurements is extracted by measuring 10 points on each wafer.

**Figure 15 sensors-20-00287-f015:**
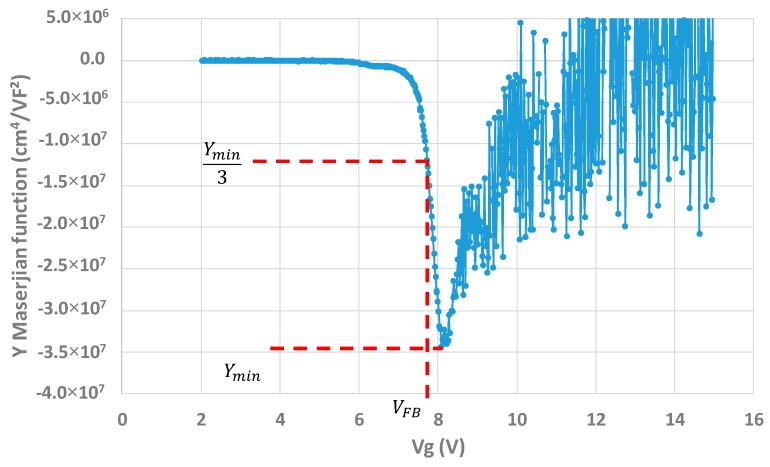
Maserjian’s Y function obtained for one capacitor test structure.

**Figure 16 sensors-20-00287-f016:**
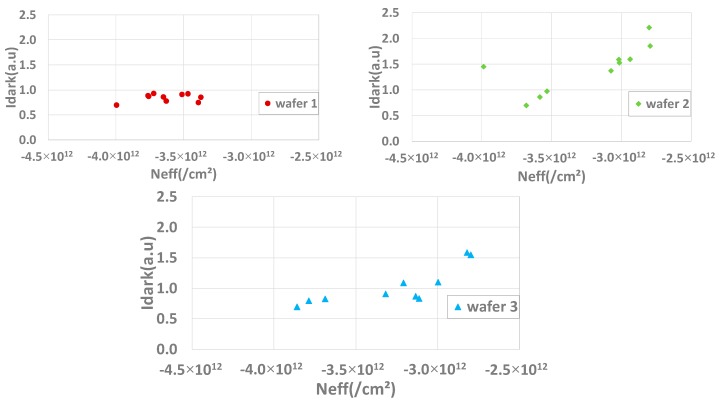
Scatter plots representing Idark as a function of N_EFF_ extracted with the Maserjian’s *Y* function for wafers 1, 2, and 3.

**Figure 17 sensors-20-00287-f017:**
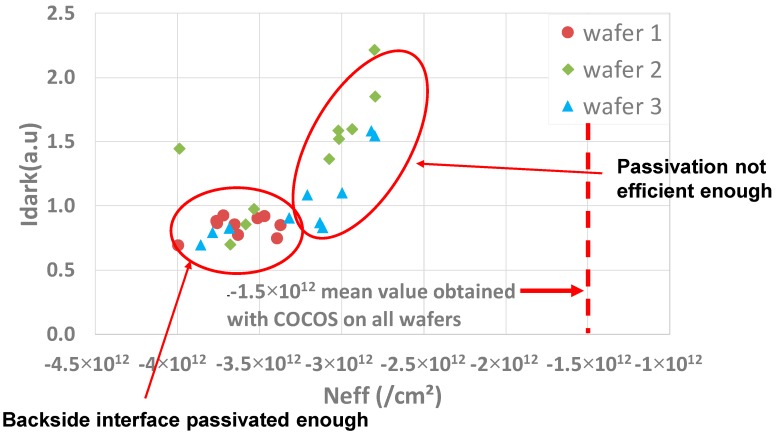
Idark(*N_EFF_*) scatter plot for the three wafers together.

**Figure 18 sensors-20-00287-f018:**
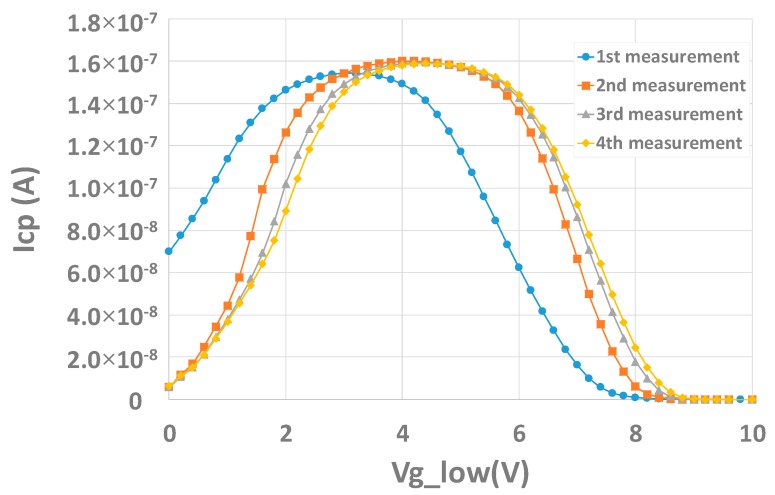
Charging effect measured on W-shield transistor structures with charge pumping measurements.

**Figure 19 sensors-20-00287-f019:**
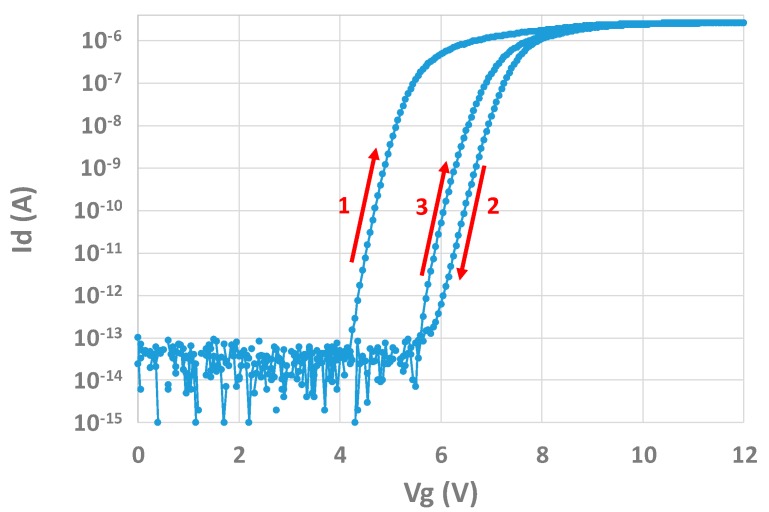
Hysteresis effect on the transistor structure.

**Figure 20 sensors-20-00287-f020:**
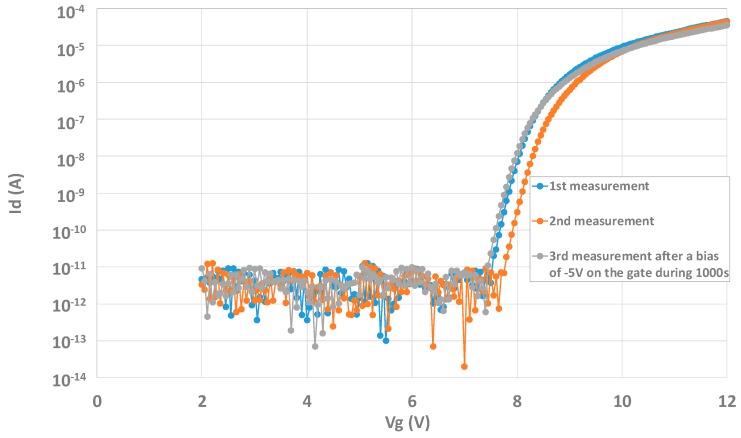
Recovery of the structure after applying a negative bias to the gate.

**Figure 21 sensors-20-00287-f021:**
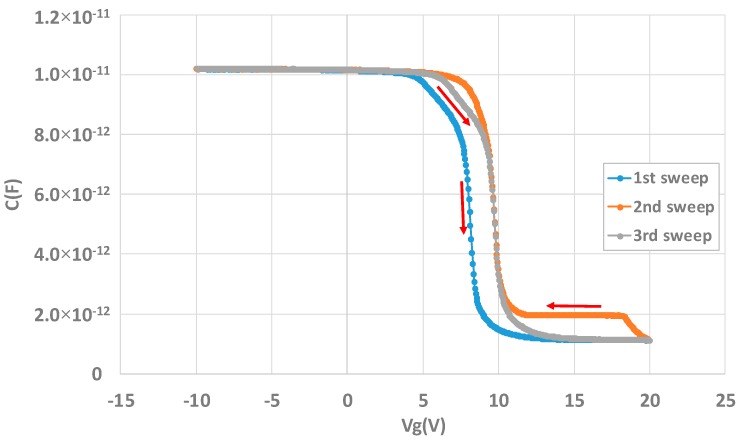
Hysteresis effect on the capacitor MOS structure.

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
