# Peer review of "Electrical Characterization of the Backside Interface on BSI Global Shutter Pixels with Tungsten-Shield Test Structures on CDTI Process"

_sensors, 2020, doi:10.3390/s20010287_

Round 1
Reviewer 1 Report
Thank you for the interesting paper. Here are some comments and suggestions that I feel would improve the manuscript.
1) Please explain in more detail how you are characterizing the dark current, reported as Idark; specifically:
How is Idark calculated? Is it an average value of multiple pixels? How many pixels, of what size? Is the pixel size used for Idark the same as the test structures used for Dit and Neff measurements? What was the CDTI bias voltage used for the measurement of Idark? How did Idark vary with the CDTI bias? Was Idark extracted for dark (covered by the W shield) pixels? What voltage of the W-back-gate was used? How does the variance and size of the tail of the dark current distribution relate to the characterization being done? Why was this not addressed in the paper? What temperature was used for Idark measurements? Did this match the temperature used for the other measurements?2) For your measurements of Dit and Neff, how many test devices are measured per die? Are these test devices physically near the pixel array?
3) Is the CV measurement show in Fig 5 used to create the results for Y shown in Fig 14? (The voltage threshold doesn't seem to match, but it's hard to tell without the raw data to do the calculation.)
4) The paper states that the CDTI bias does not affect the measurements; however, this is only shown for the case of Id vs Vg. Please show similar measurements for the charge pumping experiment (for Dit) and the CV results (for Neff) to show that the CDTI bias does not affect the results.
5) Fig 3 shows STI separating the source and drain; Fig 8 shows no STI, and a bulk connection instead. Are these two different devices? Please make the cross-section clear.
6) Can the dark current be extracted from covered pixels while varying the W-shield voltage? It seems this could demonstrate both the degree of passivation and the density of interface states directly.
7) It would be interesting to compare the measured Dit versus what is found at other interfaces in the process.
8) What is the sensitivity of the measurements to temperature? How was the temperature controlled? It would also be interesting to see the results of Idark, Dit and Neff versus temperature.
Reviewer 2 Report
Please consider improving figure 1 and the details beyond itwhat is pixel size - ( even if it referemce it make sense to have a few words on pixle seze and structure) a few words on CDTI integreation scheme / maybe a real x-section of the pixel would help about the cumulative plot - please make sure in the text it is over a wafer .. please add afew words on the text on what is the process change between wafer 1 , wafe2 and wafer 3. I think it can strengthen the paper signifcantly ( if some of the details are confidential it is OK .. not to mention them) About Figure 4 -- > I think it will make sense to have a small drawing on how you bias the structure ( maybe some tect will help ) Line 118 , I think you should re-prase to make sure how you distingush effects that are coming from long gates and temporal effects general comments on charging : can you explain in more detail hwo you refresh the device in the time related measurme ( described in the interface charecterization)
Round 2
Reviewer 1 Report
Thank you for your complete responses & modifications to the paper, you've addressed all of my questions.
Reviewer 2 Report
Dear Authors,
Most my proposal are accepted . I do understand why you choose not to accept the others.
Thank you very much.